# The Impact of Implementing an Exergame Program on the Level of Reaction Time Optimization in Handball, Volleyball, and Basketball Players

**DOI:** 10.3390/ijerph19095598

**Published:** 2022-05-05

**Authors:** Dana Badau, Adela Badau, Carmen Ene-Voiculescu, Alin Larion, Virgil Ene-Voiculescu, Ion Mihaila, Julien Leonard Fleancu, Virgil Tudor, Corina Tifrea, Adrian Sebastian Cotovanu, Alexandru Abramiuc

**Affiliations:** 1Petru Maior Faculty of Sciences and Letters, George Emil Palade University of Medicine, Pharmacy, Sciences and Technology, 540142 Targu Mures, Romania; dana.badau@umfst.ro or; 2Interdisciplinary Doctoral School, Transilvania University of Brasov, 500036 Brasov, Romania; 3Faculty of Physical Education and Sport, Ovidius University of Constanta, 900470 Constanta, Romania; alin.larion@univ-ovidius.ro (A.L.); alexandru_abramiuc@yahoo.com (A.A.); 4Mircea cel Batran Naval Academy, Naval Tactics and Armament Department, 900218 Constanta, Romania; virgil.ene@anmb.ro or; 5Faculty of Sciences, Physical Education and Informatics, University of Pitesti, 110040 Pitesti, Romania or ion.mihaila@upit.ro (I.M.); or leonef2002@yahoo.com (J.L.F.); 6Faculty of Physical Education and Sports, National University of Physical Education and Sports, 060057 Bucharest, Romania; virgiltro@yahoo.com (V.T.); c_tifrea@yahoo.com (C.T.); 7Faculty of Medicine, George Emil Palade University of Medicine, Pharmacy, Sciences and Technology, 540142 Targu Mures, Romania; adrian.cotovanu@umfst.ro

**Keywords:** human reaction times, exergames, simple motor reactions, recognition reactions, cognitive reactions, sports games, handball, basketball, volleyball

## Abstract

The main aim of the present study was to implement an exergame program that uses Fitlight technology to identify the impact on motor, recognition, and cognitive reaction times in junior athletes practicing team sports: basketball, handball, and volleyball. The second aim was to identifying differences in progress of the three types of reaction time between female and male players through computerized tests. The study included 360 subjects for basketball, 130 athletes of which were 68 male subjects and 62 female subjects; for handball, 124 athletes of which 64 were male subjects and 60 female athletes; for volleyball, 106 athletes of which 48 male were subjects and 48 female athletes. Characteristics of the experimental players: average age ± SD 13.60 ± 1.07; average sports experience ± SD 6.24 ± 0.92. The research included an initial and a final test between which a program of exergames was implemented over a period of 3 months focused on optimizing human reaction times. The evaluation of the reaction times was carried out through three computer games, the results being processed in SPSS 22. The relevant results of the research: for the simple motor reaction time (MSRT), the greatest progress between tests was the volleyball group, and for women, it was the basketball group; for the recognition reaction time (RRT), the male handball group and the female basketball group recorded the greatest progress; for the cognitive reactive time (CRT), the greatest progress was achieved by the male and female volleyball players. In all tests, the progress of the female basketball, handball, and volleyball players showed superior progress to similar male players. The results of the research highlighted the effectiveness of the experimental exergame program by using Fitlight technology in optimizing human reaction times in junior team-game athletes. Using computer games to evaluate reaction times allowed us to differentiate the evaluation on the types of human reactions under both standardized conditions but also under conditions of efficiency and attractiveness.

## 1. Introduction

Team games include a multitude of motor skills whose effectiveness is dependent on the reaction time to various stimuli. Time and the environmental and situational context of motor action in sports games influence the body’s reaction from a motor and cognitive point of view. The manifestation of athletes from a technical and tactical point of view is conditioned by the capacity of attention, anticipation, and decision making, which has a major impact in performing the technical skills specific to sports games and implicitly in improving the sports performance. The interrelation between cognitive and motor skills influences the way and speed with which athletes make decisions based on which they order and manage their motor skills in order to make them more efficient [1,2,3].

The reaction time to visual stimuli is unanimously considered as the time interval between the appearance of the stimulus and the initiation of a response of different typologies: motor, cognitive, and recognition [4,5]. Studies have shown that the reaction time is dependent on a number of factors, including the nature of the stimulus, duration of the stimulus application, the intensity of the stimulus, the afferent and efferent transmission rate of nervous influx, the processing time dependent on the complexity of the task, the size of the muscle group or segment that performs the task, etc. [6,7,8,9]. Simple motor reaction time (MSRT) is a motor reaction to kinesthetic, visual, auditory, or verbal stimuli. Recognition reaction time (RTD) is based on the cognitive processes through which the most appropriate responses to complex stimuli are chosen, and the response is dependent on the type and nature of the stimuli. Cognitive reaction time (CRT) is based on decoding, analyzing, associating and applying stimulus-specific information in relation to situational context, and cognitive complexity.

Research on reaction time in sports is extensive, but computer games have been used only in a few studies to evaluate motor, cognitive, and recognition reaction components although the trend of computerized sports technology is dynamic and current [9,10,11,12,13]. We consider the use of computer games to identify different types of reaction times to be a novel aspect in the context of increasing research in which information technology is used to prepare and evaluate different aspects of motor capacity in conjunction with cognitive capacity. Exergames facilitate the connection between sports training and the virtual environment in order to optimize physical and technical performance [14,15,16,17]. The use of exergames in sports activity has proven its usefulness through numerous studies aimed at both improving physical fitness components [18,19,20,21,22], proprioception [23,24,25], as well as the efficiency of specific motor and technical skills in individual sports [26,27,28] and in sports games, too [29,30,31,32].

Studies on the reaction time of team-sports players are relatively numerous; most studies analyzing the motor reactions of only one type of team sport in relation to specific efficiency factors, but very few compare three sports games and three types of human reactions, as we aimed to do in the current study. The main aim of the present study was to implement an exergame program that uses Fitlight technology to identify the impact on motor, recognition, and cognitive reaction times in junior athletes practicing team sports: basketball, handball, and volleyball. The second aim was to identifying differences in progress of the three types of reaction time between female and male players through computerized tests.

## 2. Materials and Methods

### 2.1. Study Design

The research was conducted between September and November 2021. The dominant hand of the study subjects was established by recording the hand with which the mouse is usually handled. The research was structured as follows: between 6–10 September 2021, the initial testing (Ti) was applied; between 13–20 November 2021 (12 weeks with 3 training sessions, 30 min per session), the exergame program was implemented; and during 22–26 November 2021, the final testing (Tf) was performed.

This research was approved on 7 June 2021 by the Review Board of the Physical Education and Sports Program of George Emil Palade University of Medicine, Pharmacy, Science, and Technology of Targu Mures, Romania. All authors of this article contributed equally; all authors have an equal contribution with the first author, too.

In our research, we applied three types of tests to measure:–Simple motor reaction time (MSRT) through the Human Benchmark test [33];–Recognition reaction time (RRT) via “Hit the Dots” [34];–Cognitive reaction time (CRT) using Part B of the Trail-making Test (TMT) [9,35,36,37,38].

The tests were organized and conducted in sessions under standardized conditions under the supervision of the authors and coaches who were trained the experimental protocol in advance. After the informed consent, the participants performed the three tests in the following order: the Human Benchmark test, the “Hit the Dots” (HD), and the TMT Part B.

Study participants were instructed to perform the tasks for the reaction time assessment test as quickly as possible. For this study, the best results in two attempts were considered for each of the three reaction time evaluation tests. Before the start of the actual test session, each subject had the opportunity to take a test in order to get used to the tests, the mouse, and the application on the phone. The initial and final testing protocol included two evaluation sessions with a 30 min break. The order of the tests was the same for the initial and final testing: the Human Benchmark test for MSRT, the Hit test for RRT, and the TMT Part B test for CRT assessment. For the Human Benchmark test and the “Hit the Dots” test, we used Lenovo computers, a production of the same generation; the HP mouse was used for all tests and players under identical conditions; the TMT Part B test was done by downloading the application on Samsung S9 phones.

### 2.2. Participants

This cross-sectional study included a total of 360 athletes. Depending on the sports game practiced, the distribution of the study subjects was as follows: for basketball, 130 athletes, of which 68 were male subjects and 62 female subjects; for handball, 124 athletes, of which 64 were male subjects and 60 female athletes; for volleyball, 106 athletes, of which 48 were male subjects, and 48 female athletes. Characteristics of the experimental players: average age ± SD 13.60 ± 1.07; average sports experience ± SD 6.24 ± 0.92.

The criteria for including these subjects in the study: active athletes practicing selected sports for research, in good health, and 13–14 years of age. Exclusion criteria: interruption of sports activity for more than one month due to injuries or the pandemic situation, failure to complete the tests, and failure to complete the experimental program of exergames. Participation in the test was voluntary based on the informed consent of each participant, and the study follows the principles of the Declaration of Helsinki.

### 2.3. Experimental Exergame Program of Study

In the research, a 12-week program for 3 times a week, with 30 min per training session, was applied only in the experimental group. The program included exergame exercises using Fitlight technology [39], cards with different encryption (letters, figures, numbers, etc.), which facilitated us to design a set of 24 exercises to improve the ability to react to visual stimuli. The motivation for using Fitlight technology is the good level of reliability in the accessibility of using a variable number of LED wireless lights connected to the Fitlight computer platform that can be arranged in different shapes and on different horizontal and vertical surfaces. Fitlight technology has been applied in numerous studies to highlight the effectiveness of improving reaction speed and cognitive processes with applicability in sports.

The content of the experimental exergame program was structured in three subprograms: eight exercises for optimizing the motor reaction time: MSRT subprogram; eight exercises for developing the recognition reaction time: RRT subprogram; and eight exercises for the cognitive reaction time: CRT subprogram. For each training session, two exercises from the three subprograms were scheduled. The experimental program of exergames was applied identically to all three experimental groups of athletes practicing sports games: handball, volleyball, and basketball.

### 2.4. Measures

The research subjects had at their disposal two attempts of each test that used frontal-disposed visual stimuli, and the highest achieved value was taken into account for the study.
–Simple motor reaction time (MSRT) was assessed by the Human Benchmark computerized test [33]. The Human Benchmark test consisted of 5 executions/2 attempts, and the arithmetic average of the times achieved for each attempt was recorded, taking into account the best value.–Recognition reaction time (RRT) was assessed by the “Hit the Dots” reaction test [34]. The “Hit the Dots” computer game test included 2 attempts to get the best result and consisted of clicking as many black dots as possible from the maximum 60 points of the test, which are arranged in 6 lines with 10 circles each, within 30 s. The Hit the Dots test was designed and validated by the University of Washington and allows for standardized and easy use.–The cognitive reaction time (CRT) assessed by Trail-making Test (TMT) Part B [35], which is dedicated to measuring cognitive flexibility. TMT Part B consists of 25 circles that are arranged on the screen randomly, and the subjects must click and associate the numbers with the letters in ascending order according to the pattern 1—A; 2—B...12—L, 13—M, etc.

For the “Hit the Dots” test, the number of the achieved points was taken into account, for the Human Benchmark and TMT part B tests, the unit of measurement was in seconds (sec), with the best time being calculated.

### 2.5. Statistical Analysis

The data were processed using IBM-SPSS 22. The main statistical indicators were: average (X), standard deviation (SD), and average difference between tests (∆XT). For the parametric comparison of two groups, we used the Student’s *t*-test (t), the confidence interval with two levels, lower and upper (95% CI), and d effect size. The interpretation of the Cohen’s d effect size was: 0.1–0.2 small, 0.3–0.5 medium, 0.5–0.8 large, and over 0.8 very large. The difference between the means of the three independent groups were analyses with ANOVA. The value of the statistical reference significance for this study was selected at *p* < 0.05.

## 3. Results

In Table 1, regarding the analysis of the simple reaction time (MSRT), it can be found that in all three categories of sports included in the study, namely basketball, handball, and volleyball, both in the female and the male groups, there was a statistically significant progress between the final and the initial testing. Analyzing the results between tests, we can highlight the fact that in all tests of the study, the female players progressed more than the male players in all sports as follows: with 4.03 s in basketball, with 1.56 s in handball, and with 4.68 s in volleyball. The analysis of the differences of the arithmetic average recorded between the two tests of the study reveals that they were within the limits of the confidence coefficient (95%). The analysis of the Cohen’s d statistical parameters of our research showed that the results of the Human Benchmark tests were between 0.459–0.582 for male samples and 0.530–0.640 for female samples, which is considered between medium and large effect sizes for all samples of the three sports games.

For male players, the greatest progress between tests was recorded by volleyball players with 14.85 s, followed by those playing basketball with 9.77 s, and the least progress was documented as 8.79 s. For the female players, the greatest progress between tests was recorded by the volleyball players with 19.53 s, followed by 13.80 s, and the least progress was documented as 10.35 s.

The analysis of the results recorded in Table 2 in the “Hit the Dots” test highlights some relevant aspects regarding the improvement of the RRT time as a result of the implementation of the experimental program of exergames. Comparing the results between the initial and final testing for the male players, we can find a progress of the recognition reaction time of 11.66 points in basketball, 12.17 points in handball, and 11.60 points in volleyball; and in the case of the female players, we can find the same progress of 12.59 points in basketball, 12.45 points in handball, and 11.87 points in volleyball. Depending on the progress made, the ranking of the male groups according to the team sport practiced is as follows: handball, volleyball, and basketball. For the female groups, the ranking is basketball, handball, and volleyball. All the recorded average differences were between the lower and upper limits for 95% C.I. The analysis of the Cohen’s d statistical parameters of our research showed that the results of the “Hit the Dots” tests were between 0.469–0.611 for male samples and 0.546–0.675 for female samples, which considered as between medium and large effect sizes for all samples of the three sports games.

Comparing the results between the male and female players in the RRT test, we can find the following differences: in basketball with 1.07 points in favor of the female players, in handball with 0.28 points in favor of the female players, and in volleyball with 0.27 points also in favor of the female players. In Table 2, it is found that the female players progressed more compared to the male ones; also, all the results recorded by both types of players were statistically significant for *p* < 0.05.

According to Table 3, in all results for TMT Part B, all players and all tests were statistically significant, with the *t* values recorded in the study being lower than the selected significance threshold of *p* < 0.05. Analyzing the differences of the arithmetic averages recorded between the two tests, we can find that they fall between the limits of 95% C.I. The male players recorded the following ranking according to the value of the difference of the arithmetic averages between the tests: volleyball with 3.66 s, basketball 2.48 s, and handball 1.97 s; in the case of the female players, their evaluation according to the progress made between the research tests is: volleyball 3.80 s, basketball with 3.18 s, and in handball with 3.13 s. Comparing the progress of the research between the players according to gender, it results that all the female players registered superior progress compared to the male ones in the TMT test part B as follows: in basketball with 0.70 s, in handball with 1.16 s, and in volleyball with 0.14 s. The analysis of the Cohen’s d statistical parameters showed that the results of the TMT Part B tests were between 0.438–0.628 for male samples and 0.539–0.719 for female samples, which is considered between medium and large effect sizes for all samples for volleyball, handball, and basketball.

The use of ANOVA analysis of variance allowed us to identify the differences that were statistically significant between the averages for the basketball, handball, and volleyball groups in all three human reaction assessment tests with one exception. The results in Table 4 confirm significant differences in the initial and final tests of the male samples in the tests: Human Benchmark, Hit the Dots, and TMT Part B. For female groups, the differences were statistically significant for all tests: Human Benchmark and TMT Part B but also in the Hit test for both the initial and final test, too.

## 4. Discussion

Our study aimed to implement an exergame program that uses Fitlight technology to identify the impact on motor, recognition, and cognitive reaction times in junior athletes practicing team sports, namely basketball, handball, and volleyball. The second aim focused to identifying differences in progress of the three types of reaction time between female and male players through computerized tests. The specific results of our study regarded the impact of the experimental program of exergames on three types of reaction times, namely simple motor reaction time, recognition reaction time, and cognitive reaction time, and we consider that they facilitate the expansion of the level of knowledge about sports performance and especially about the human reaction capacity to visual stimuli. The results of the research contribute to the expansion of knowledge on the evaluation of reaction times to visual stimuli specific to the three computer games tests as well as the experimental program that used Fitlight technology that included LED wireless lights.

In all tests of our study conducted through standardized computer games, significant progress was made between tests in favor of final testing, which demonstrates the effectiveness of the experimental program implemented by exergames using Fitlight technology, the contents of which focused on improving three categories of human reaction times in junior athletes in three team sports. According to the progress made between the research tests, we found that the female basketball players recorded the highest progress in MSRT and RRT, the volleyball one in CRT, and in the case of male players, the best results were recorded by volleyball in MSRT and CRT and handball in RRT. These results highlight the differences between the female and male players in terms of the progress achieved as a result of the implementation of an experimental program of exergames in relation to the specifics of the team sport practiced and the gender particularities. Our results confirm previous studies that have highlighted gender differences in correlation with the typology of the practiced sport [40,41,42,43].

Analyzing the results of the study, for all types of reaction times, it was found that female players of basketball, handball, and volleyball progressed more compared to similar male players included in the study. We consider that the superior progress of the female players compared to the male ones is due to a better ability to focus in the practice of the experimental program and when performing computer games tests, with these being associated with the peculiarities of motor and mental development at the age of 13–14. We believe that our results are in agreement with previous studies specific to sports activity that highlight differences between genders correlated with age characteristics specific to puberty [44,45,46].

Analyzing the specialized nature, there are numerous studies that approach the optimization of human reaction time [47,48,49] in team games, namely handball [50,51,52,53], basketball [54,55], volleyball [56,57,58], and other team games [59,60]. We also identified several studies that address the interrelation between different types of human reactions, especially between individual sports [9,61,62,63], and very few between team sports [64,65,66,67]. A number of studies demonstrated that the reaction times of male players, especially in adults, are better than in female ones due to the fact that motor responses are corroborated with muscle contraction capacity, which is also emphasized by our study in terms of average results in initial and final tests [68,69,70].

Due to the method of evaluation through computer games, although the results in the initial and final tests of the female players were lower compared to the male ones, the progress made as a result of the implementation of the experimental program, however, was superior in the female players compared to the male ones. We consider that the superior progress of the female players is due to a capacity of concentration and reaction to the tasks of the evaluation tests, which is superior to the male players; of course, this fact is also explainable in terms of gender peculiarities for the age group of 13–14 years [71,72,73,74].

Exergame programs aimed at improving human reaction times can be adapted and implemented in recreational, prophylactic, and sport activities for different age groups due to their varied possibilities for practice, organization, and attractiveness [75,76,77,78,79]. The modernization of the training of team sports players must be at the center of the specialists’ concerns in order to optimize the physical and technical performance potential that can be solved by including information technologies, exergames, etc.

Strengths. The present study included the design and implementation of an experimental program of exergames using Fitlight technology and which focused on the development of three types of human reactions, namely motor, recognition, and cognitive. The study combined the exercises of the subprograms of the experimental program, which facilitated the optimization of human reaction capacity, and was conducted on a number of three categories of team sports games, namely basketball, handball, volleyball. Another strength is the large number of junior athletes involved in the study as well as the use as testing tools of computer standardized games, which have proven their validity, attractiveness, and reduced time and resources. The results of our study facilitate the generalization of the experiential program of exergames to other team sports and to other age groups.

Limitations. The involvement of athletes from only three team sports and not from several team sports and the non-correlation of the results of the tests through computer games with the results of the athletes in the motor tests represent limitations as well as the limited duration of implementation of the experimental program of exergames.

## 5. Conclusions

The results of our study confirmed that the implemented exergame program that uses Fitlight technology had a significant impact on motor, recognition, and cognitive reaction times in junior athletes practicing the team sports of basketball, handball, and volleyball. The study showed significant progress both between the initial and the final testing as well as between the female and male players. Depending on the progress made between the research tests, the analysis of male players according to the practiced sport, in descending order, is as follows: MRST volleyball in first place, followed by basketball and handball; for RRT handball in the first place, followed by basketball and volleyball; and for CRT volleyball in on the first place, followed by basketball and handball. The ranking in descending order of the female players according to the progress made in the study is as follows: for MRST basketball in first place, followed by handball and volleyball; for RRT basketball in first place, followed by handball and volleyball; and for CRT volleyball in first place, followed by basketball and handball. Analyzing the results of the study, for all types of reaction times, it was found that female players of basketball, handball, and volleyball progressed more compared to similar male players included in the study.

## Figures and Tables

**Table 1 ijerph-19-05598-t001:** Statistical analyses of the results of the MSRT for the team sports.

Test	Team Sports	Gender	TiX ± SD	TfX ± SD	∆XTs	95% C.I. Lower, Upper	*t*	*p*	d
Human Benchmark (sec)	Basketball	M	264.10 ± 21.80	254.32 ± 21.33	9.77	18.91; 0.64	2.13	0.036	0.550
		F	266.03 ± 6.41	252.22 ± 7.273	13.80	20.07; 7.53	4.40	0.000	0.640
	Handball	M	267.78 ± 23.99	258.98 ± 20.18	8.79	−16.17; 1.42	2.38	0.020	0.459
		F	269.35 ± 38.55	259.00 ± 31.76	10.35	−20.46; 0.23	2.04	0.045	0.530
	Volleyball	M	287.39 ± 33.64	302.25 ± 40.24	14.85	1.57; 28.13	2.25	0.029	0.577
		F	286.33 ± 39.60	305.86 ± 38.26	19.53	11.00; 28.06	4.58	0.000	0.582

Ti, initial test; TF, final test; X, arithmetic average; SD, standard deviation; *t*, Student’s *t*-test, XT, average differences between tests; 95% C.I., interval of confidence with lower and upper levels; d, effect size.

**Table 2 ijerph-19-05598-t002:** Statistical analyses of the results of the RRT for the team sports.

Test	Team Sports	Gender	TiX ± SD	TfX ± SD	∆XTs	95% C.I. Lower, Upper	*t*	*p*	d
Hit the Dots (points)	Basketball	M	19.32 ± 4.37	30.98 ± 3.63	11.66	10.95; 12.37	32.707	0.000	0.469
		F	17.40 ± 3.99	30.00 ± 3.01	12.59	11.80; 13.38	31.761	0.000	0.583
	Handball	M	18.07 ± 4.35	30.25 ± 3.17	12.17	10.81; 13.52	17.964	0.000	0.611
		F	16.33 ± 4.79	28.78 ± 3.49	12.45	11.15; 13.74	19,275	0.000	0.675
	Volleyball	M	17.14 ± 4.60	28.75 ± 1.98	11.60	10.16; 13.03	16.271	0.000	0.491
		F	15.98 ± 3.01	27.86 ± 2.10	11.87	10.82; 12.93	22.542	0.000	0.546

Ti, initial test, TF—final test; X, arithmetic average; SD, standard deviation; *t*, Student’s *t*-test; XT, average differences between tests; 95% C.I., interval of confidence with lower and upper levels; d, effect size.

**Table 3 ijerph-19-05598-t003:** Statistical analyses of the results of the CRT for the team sports.

Test	Team Sports	Gender	TiX ± SD	TfX ± SD	∆XTs	95% C.I. Lower, Upper	*t*	*p*	d
TMT Part B (sec)	Basketball	M	61.31 ± 10.26	58.82 ± 11.07	2.48	4.21; 0.75	2.873	0.005	0.518
		F	64.47 ± 6.98	61.28 ± 7.92	3.18	4.40; 1.97	5.250	0.000	0.572
	Handball	M	59.58 ± 6.50	57.60 ± 7.47	1.97	3.59; 0.35	2.432	0.018	0.438
		F	58.40 ± 8.25	55.26 ± 8.86	3.13	5.47; 0.80	2.688	0.009	0.539
	Volleyball	M	68.67 ± 10.01	65.00 ± 11.71	3.66	6.84; 0.48	1.460	0.025	0.628
		F	69.29 ± 10.93	65.49 ± 6.73	3.80	6.14; 1.46	3.250	0.002	0.719

Ti, initial test; TF, final test; X, arithmetic average; SD, standard deviation; *t*, Student’s *t*-test; XT, average differences between tests; 95% C.I., interval of confidence with lower and upper levels.

**Table 4 ijerph-19-05598-t004:** ANOVA (Analysis of Variance) between basketball, handball, and volleyball players.

Test Types	Gender	Tests	∑	df	Ms	F	*p*
Human Benchmark (sec)	M	Ti	15,737.519	2	7868.759	8.866	0.000
		Tf	74,123.861	2	37,061.931	39.144	0.000
	F	Ti	15,429.205	2	7714.602	6.181	0.003
		Tf	104,942.401	2	52,471.200	49.120	0.000
Hit the Dots (points)	M	Ti	138.329	2	69.165	3.520	0.032
		Tf	142.076	2	71.038	7.362	0.001
	F	Ti	66.342	2	33.171	2.061	0.030
		Tf	138.315	2	69.157	8.005	0.000
TMT Part B (sec)	M	Ti	2503.289	2	1251.645	15.290	0.000
		Tf	1631.144	2	815.572	7.909	0.001
	F	Ti	3519.995	2	1759.998	22.544	0.000
		Tf	3125.464	2	1562.732	25.031	0.000

M, male; F, female; ∑, sum of squares; df, degrees of freedom; MS, mean square; F, F test value; *p*, probability level.

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
