# Peer review of "The Impact of Implementing an Exergame Program on the Level of Reaction Time Optimization in Handball, Volleyball, and Basketball Players"

_ijerph, 2022, doi:10.3390/ijerph19095598_

Round 1

Reviewer 1 Report

The paper presents substantial research. The study is carefully prepared, conducted and evaluated. What I miss at several places are justifications for design decisions. For example,  Fitlight technology is used. How exactly does it work? Are there alternatives? Why they were not chosen? What is the "Hit the Donuts test"? And why it is particularly useful here? The statistical evaluation focuses strongly on statistical significance which is sometimes critizised. What about effect sizes? The paper is not very nice to read. It contains many statistical information but not a single image. It is about "visual stimuli", but they are not shown. Even bullet lists would help to emphasize some information, for example, in Sect. 3.1. there are three test types mentioned - that could be a list.

In the Introduction, I'm not sure what "Stimulation time" means. Morning, evening, or stimulation duration?

There is a section 2 "Materials and Methods" and then follows immediately 3.1 "Study design". 

Author Response

Dear Reviewer

I really appreciate your recommendations. We tried, based on all the recommendations to improve and correct the article. I take into consideration the recommendations and I improved the sections of article.

Thank you for your support and help.

Reviewer 2 Report

The manuscript is interesting and original, dealing with sensitive themes . It presents various case studies, so increasing the scientific importance of the research.

The coherence and rigor of methodological application are relevant; clarity and exposure style highlight organization of the material.

Just to broaden the scope of the research It is advisable to cite the following bibliographical references:

Lugeri, F.R.; Farabollini, P. Discovering the Landscape by Cycling: A Geo-Touristic Experience through Italian Badlands. Geosciences 2018, 8, 291. https://doi.org/10.3390/geosciences8080291

Eichberg, H. (1998). Body Cultures: Essays on Sport, Space, and Identity (J. Bale & C. Philo, Eds.). London: Routledge.

Ewert, A., McCormick, B., & Voight, A. (2001). Outdoor Experiential Therapies: Implications

for         TR          Practice.                Therapeutic                Recreation                Journal,                35(2).        

http://js.sagamorepub.com/trj/article/view/1068

Author Response

(The authors gave the same response as above.)

Reviewer 3 Report

Improving motor, recognition and cognitive reaction times in junior athletes practicing team sports is important, but the purpose of this paper is to identify the difference between basketball, handball and volleyball athletes after 12 week exergame training program, that confuses the readers. In my opinion, basketball, volleyball and handball are three different kinds of ball games, and reaction time differences in progress between female and male players through computerized tests are always existent, so the meaning of this design is not obvious. What kind of performance effects will be brought to ball games after the improvement of reaction time with 12 week interaction, that is meaningful to this study.

Author Response

(The authors gave the same response as above.)

Round 2

Reviewer 3 Report

Thank you for your hard work. The changed purpose of the  study is more suitable for the study design, because motor, recognition and cognitive reaction times are very important in junior athletes practicing team sports.